# Pref-GUIDE: Continual Policy Learning from Real-Time Human Feedback via Preference-Based Learning

**Zhengran Ji**                    *zhengran.ji@duke.edu*
*Duke University*

**Boyuan Chen**                    *boyuan.chen@duke.edu*
*Duke University*

**Project Website:** `http://generalroboticslab.com/Pref-GUIDE`

**Reviewed on OpenReview:** `https://openreview.net/forum?id=dWGUwidXDm`

## Abstract

Training reinforcement learning agents with human feedback is crucial when task objectives are difficult to specify through dense reward functions. While prior methods rely on offline trajectory comparisons to elicit human preferences, such data is unavailable in online learning scenarios where agents must adapt on the fly. Recent approaches address this by collecting real-time scalar feedback to guide agent behavior and train reward models for continued learning after human feedback becomes unavailable. However, scalar feedback is often noisy and inconsistent, limiting the accuracy and generalization of learned rewards. We propose PREF-GUIDE, a framework that transforms real-time scalar feedback into preference-based data to improve reward model learning for continual policy training. PREF-GUIDE `Individual` mitigates temporal inconsistency by comparing agent behaviors within short windows and filtering ambiguous feedback. PREF-GUIDE `Voting` further enhances robustness by aggregating reward models across a population of users to form consensus preferences. Across three challenging environments, PREF-GUIDE significantly outperforms scalar-feedback baselines, with the `voting` variant exceeding even expert-designed dense rewards. By reframing scalar feedback as structured preferences with population feedback, PREF-GUIDE offers a scalable and principled approach for harnessing human input in online reinforcement learning.

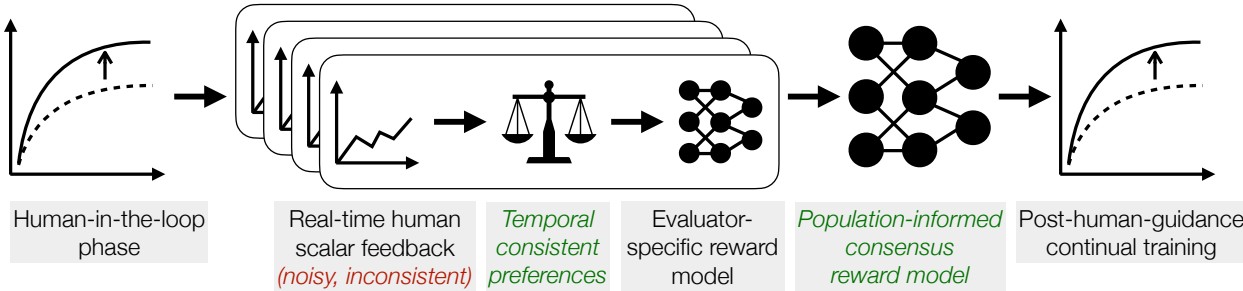

Figure 1: **Pref-GUIDE.** Real-time scalar human feedback is often inconsistent, noisy, and varies across individuals. PREF-GUIDE addresses this by (a) converting scalar feedback into local pairwise preferences to achieve temporal consistency, and (b) aggregating reward models across human evaluators to form consensus-based rewards. (c) These improvements yield more robust reward learning and enable effective continual policy training after human feedback becomes unavailable.

# 1 Introduction

Incorporating human feedback into reinforcement learning (RL) has become increasingly important for training agents in environments where task goals are difficult to formalize Christiano et al. (2017); Cao et al. (2021); Rafailov et al. (2023); Ramesh et al. (2024); Zhang et al. (2024b); Ji et al. (2025). While manually designed reward functions can work well in constrained or well-specified settings, they often fail to capture the full complexity of real-world objectives or align with human intent. In such cases, human feedback offers a more flexible and adaptive alternative for guiding agent behavior.

Two major paradigms have emerged to leverage human feedback in RL. Preference-based RL Christiano et al. (2017); Cao et al. (2021); Ouyang et al. (2022); Rafailov et al. (2023); Ramesh et al. (2024) uses offline datasets of pairwise trajectory comparisons to train reward models that reflect human judgments. These models are then used to generate reward signals during policy learning. Alternatively, real-time scalar feedback MacGlashan et al. (2017); Arumugam et al. (2019); Knox & Stone (2009); Warnell et al. (2018); Zhang et al. (2024b) allows humans to directly influence the learning process by providing moment-to-moment evaluations during agent interaction. GUIDE Zhang et al. (2024b), a recent method in this category, introduces a two-phase framework: during the `human-in-the-loop phase`, scalar feedback is combined with environment rewards to guide agent learning; in the subsequent `post-human-guidance phase`, a regression model trained on the collected human feedback continues to guide the agent after the human input is no longer available.

While effective in early demonstrations, both paradigms face critical limitations. Preference-based methods rely on rich trajectory sets and offline human labeling, making them unsuitable for real-time and interactive settings Wirth et al. (2017a). Moreover, the co-evolution of the policy and reward model can introduce instability Wirth et al. (2017a). Real-time scalar feedback, while more adaptable, suffers from two key challenges, especially in obtaining an effective reward model for `post-human-guidance phase`:

**Human feedback is inconsistent over time.** Evaluators naturally shift their expectations over the course of training Wang et al. (2022), which makes it difficult to learn stable reward functions from raw feedback. For example, in an object search task, exploratory behavior may initially be rewarded but later penalized once goal-directed behavior is expected. This temporal inconsistency introduces non-stationarity, making it difficult for a regression model to generalize feedback into a stable reward model.

**Human feedback is not always reliable.** Variability in cognitive ability, fatigue, attention lapses, and evaluation Wang et al. (2022); Wirth et al. (2017a); Zhang et al. (2024a;b) contribute to intra- and inter-evaluator noise. These inconsistencies reduce the signal-to-noise ratio in scalar feedback, ultimately degrading policy performance when learned directly.

Our goal in this study is to improve the `post-human-guidance phase` of real-time human-guided RL. While we adopt the same `human-in-the-loop phase` as GUIDE, our contributions focus on transforming and aggregating the collected human feedback to support more stable and effective continual training without requiring additional human input. To this end, we propose PREF-GUIDE (Figure 1), a framework that transforms noisy real-time scalar feedback into structured preference data to train more robust reward models. PREF-GUIDE consists of two key components:

**Pref-GUIDE `Individual`** transforms scalar human feedback from each evaluator into temporally local pairwise preferences. This design is motivated by our observation (Section 4.1) that human feedback tends to be relatively consistent within short time intervals. By comparing agent behaviors within short temporal windows, we generate multiple preference pairs from each scalar signal, which effectively reuses feedback to create more training data while maintaining temporal coherence. We further introduce a no-preference margin Lee et al. (2021b) to handle ambiguous or low-confidence feedback.

**Pref-GUIDE `Voting`** builds on this idea by aggregating individual reward models across a population of human evaluators. Since feedback quality varies across individuals due to noise or evaluator bias Wang et al. (2025); Meyers et al. (2023); von Rueden et al. (2015), we use a consensus-based relabeling strategy to extract a more robust reward signal. By combining predictions from multiple evaluator-specific models, we reduce individual noise and improve the robustness of the learned reward (Figure 1).

We evaluate Pref-GUIDE across three challenging visual RL environments Zhang et al. (2024a;b), where agents must act based on partial visual observations. Our results show that Pref-GUIDE Individual outperforms regression-based baselines when feedback quality is high, and Pref-GUIDE Voting maintains strong performance across diverse user inputs. In more complex tasks, our method even surpasses policies trained with expert-designed dense rewards, demonstrating the power of structured and population preference learning from real-time human feedback. Extensive ablation studies further validate our design choices and provide insight into their contributions.

By converting noisy and unreliable human signals into structured and population-aggregated preference data, Pref-GUIDE offers a scalable and principled framework to leverage human feedback in RL, towards more consistent improvements even after human supervision ends.

## 2 Related Work

### 2.1 Real-Time Human-in-the-Loop Reinforcement Learning

Incorporating real-time human feedback into RL has been studied through several paradigms, each grounded in different assumptions about the nature and role of feedback. One major class of methods, such as TAMER Knox & Stone (2009) and Deep TAMER Warnell et al. (2018), treats human input as a one-step reward signal, allowing agents to update their policies myopically based on predicted human evaluations. A second line of work, such as COACH MacGlashan et al. (2017) and Deep COACH Arumugam et al. (2019), interprets feedback as an advantage estimate, providing a more temporally grounded signal that can reduce the effects of inconsistent feedback. More recently, GUIDE Zhang et al. (2024b) has shown that using real-time scalar human feedback directly as a dense reward signal can provide superior performance in challenging visual RL tasks.

On the other hand, while prior approaches have shown the promise of real-time human feedback in RL, they typically assume continuous human involvement. GUIDE overcomes this assumption by training a regression reward model from the human feedback data to provide continual training signals after the human feedback is no longer available. However, there are two key limitations in GUIDE. First, GUIDE relies on point-wise scalar prediction, which limits its ability to capture the evolving nature of human evaluations. Second, GUIDE's performance strongly correlates with the individual human evaluators, where agents guided by human evaluators with stronger cognitive skills in certain aspects perform much better. To have human-guided RL algorithms agnostic to human individual differences, such correlation is undesirable.

Our work Pref-GUIDE builds on the real-time human-guided RL framework proposed in GUIDE, but addresses these limitations through two novel algorithm designs. Pref-GUIDE considers that human evaluation criteria evolve over time by grounding scalar feedback into pairwise preference labels within a short time window. Moreover, to mitigate human biases and individual differences, Pref-GUIDE aggregates individual reward models into consensus-based labels, improving the robustness of the reward model based on the feedback from population users.

### 2.2 Preference-Based Reinforcement Learning

Preference-based RL (PbRL) Wirth et al. (2017b) obtains human feedback to train RL agents by collecting pairwise trajectory comparisons and inferring a latent reward function by modeling human preferences Bradley & Terry (1952); Christiano et al. (2017). The learned reward model will be used to supervise policy learning with dense reward estimates.

A rich body of work has expanded the PbRL framework to improve data efficiency and label quality. This includes integrating human demonstrations for pretraining Ibarz et al. (2018), introducing soft labels to capture uncertainty Cao et al. (2021), leveraging relabeling and unsupervised pretraining Lee et al. (2021a), and more recently, using foundation models to automate preference labeling Wang et al. (2024); Jian et al. (2025). These methods typically operate in an offline or evolution loop setting, where the agent's policy is periodically updated based on human feedback collected over batches of offline rollouts Christiano et al. (2017); Ibarz et al. (2018); Lee et al. (2021a;b).

However, such iterative co-evolution of policy and reward model can introduce training instability, as both the learned behaviors and feedback targets shift over time Wirth et al. (2017a). Moreover, the requirement for human queries on offline datasets or parallel policy rollouts makes it unclear to apply PbRL approaches directly in real-time decision-making tasks.

Our method, PREF-GUIDE, adapts the strengths of PbRL to the real-time feedback setting. Rather than requiring humans to explicitly compare trajectories, we transform real-time scalar feedback, naturally provided in real-time settings, into temporally grounded pairwise preferences. This allows us to retain the expressivity and consistency benefits of PbRL without the overhead of repeated and offline query loops. Furthermore, while most existing PbRL work either trains evaluator-specific models or pools preference data from all evaluators without addressing inter-evaluator variability, PREF-GUIDE introduces a population-level aggregation mechanism. Through PREF-GUIDE `Voting`, we combine individual reward models into soft consensus labels, improving the robustness of the learned reward signal in continual learning against individual differences.

## 3 Preliminaries

### 3.1 Real-Time Human-Guided Reinforcement Learning

In real-time human-guided RL, a human observer provides scalar feedback while watching the agent interact with the environment. Let $\tau = (s_0, a_0, s_1, a_1, \ldots, s_T, a_T) \in (\mathcal{S} \times \mathcal{A})^T$ denote a trajectory consisting of states and actions, where $s_t \in \mathcal{S}$ and $a_t \in \mathcal{A}$ represent the state and action at time step $t$. The human provides a scalar evaluation $f \in \mathbb{R}$ for the trajectory $\tau$, reflecting their assessment of the agent's behavior to the task objective.

Different methods adopt different formats for scalar feedback. For instance, in TAMER Knox & Stone (2009), $f$ is a discrete value selected from $\{-1, 0, 1\}$ to represent negative, neutral, or positive responses. GUIDE Zhang et al. (2024b) further improves this by using a continuous value in the range $[-1, 1]$ to capture more nuanced human feedback on a spectrum from negative to positive.

**GUIDE Zhang et al. (2024b) algorithm.** GUIDE is a recent framework in real-time human-guided RL to enable higher performance and faster convergence of RL agents using real-time human feedback. There are two main phases in GUIDE:

**Human-in-the-loop phase:** The continuous value feedback signals from human evaluators are directly combined with sparse terminal rewards to guide the policy training. Meanwhile, these feedback signals are stored as a dataset $\mathcal{D}_{\text{real-time}} = \{(\tau_i, f_i)\}_{i=1}^N$ for training a reward model with a regression loss.

**Post-human-guidance phase:** Once human evaluators exit the loop, the reward model will be used to estimate human feedback to continue guiding the training.

### 3.2 Preference-Based Reinforcement Learning

Preference-based reinforcement learning (PbRL) is commonly used when pairwise preference labels are available, either from existing offline datasets or from parallel rollouts that allow trajectory comparisons. Rather than relying on scalar feedback values, PbRL uses human judgments over pairs of trajectories to learn a reward model. This reward model estimates latent reward values consistent with the provided preferences and can be integrated into standard reinforcement learning pipelines to guide policy optimization.

In PbRL, a preference dataset $\mathcal{D}_{\text{pref}} = \{(\tau_i^A, \tau_i^B, y_i)\}_{i=1}^M$ is collected by asking humans to compare pairs of trajectories. Each label $y_i$ indicates which trajectory is preferred: $y_i = 1$ if $\tau_i^A \succ \tau_i^B$, $y_i = 0$ if $\tau_i^B \succ \tau_i^A$, and $y_i = 0.5$ if the annotator expresses no preference.

We assume preferences are generated according to a latent reward function $r_\theta : (\mathcal{S} \times \mathcal{A})^T \to \mathbb{R}$, parameterized by $\theta$. Following the Bradley-Terry model Bradley & Terry (1952), the probability that trajectory $\tau^A$ is

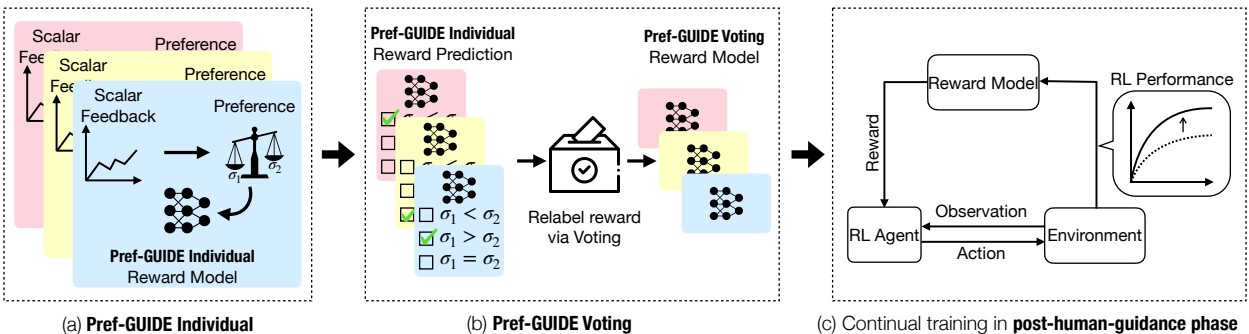

Figure 2: **Method Overview.** (a) PREF-GUIDE `Individual` converts real-time scalar feedback from each human evaluator into a localized preference dataset, then trains evaluator-specific reward models. (b) PREF-GUIDE `Voting` aggregates predictions from these individual models to relabel trajectory pairs through consensus voting, providing a population-informed preference dataset and a robust reward model. (c) The aggregated reward model is used to guide RL training during the `post-human-guidance phase`.

preferred over $\tau^B$ is the softmax likelihood of the reward model predictions:

$$P(\tau^A \succ \tau^B) = \frac{\exp(r_\theta(\tau^A))}{\exp(r_\theta(\tau^A)) + \exp(r_\theta(\tau^B))}.$$

The reward model is trained by minimizing the binary cross-entropy loss over the set of preference pairs:

$$\mathcal{L}_{\text{pref}}(\theta) = -\sum_{i=1}^{M} \left[ y_i \log P(\tau_i^A \succ \tau_i^B) + (1 - y_i) \log P(\tau_i^B \succ \tau_i^A) \right].$$

Once learned, the reward model $r_\theta$ can be used to generate dense reward estimates for new trajectories, enabling the use of standard RL algorithms for policy optimization.

## 4 Method

Our method, termed PREF-GUIDE, is designed to improve the `post-human-guidance phase` of real-time human-guided RL. While recent methods have focused on improving the initial phase where human evaluators provide real-time scalar feedback, our focus is on what happens after the human is no longer available. This phase is critical for scaling real-time human-guided RL, as continuous human supervision, though intuitive and effective, is costly, time-consuming, and ultimately unsustainable. An overview of PREF-GUIDE is shown in Figure 2.

To address this, PREF-GUIDE introduces two key improvements aimed at making better use of the limited human feedback collected during training: (1) converting scalar feedback into temporally local pairwise preferences to enhance consistency and data efficiency, and (2) aggregating feedback across multiple human evaluators to reduce individual noise and bias. These improvements are realized through two complementary modules: PREF-GUIDE `Individual`, which transforms individual feedback streams into structured preference data, and PREF-GUIDE `Voting`, which combines reward models from multiple evaluators to generate robust, consensus-based labels. We describe each component in detail in the following sections.

**Human guidance data.** Our dataset $\mathcal{D}_{\text{real-time}}$ comes from GUIDE Zhang et al. (2024b), which contains interactions from 50 human evaluators across three challenging visual RL environments: Bowling, Find Treasure, and Hide and Seek 1v1. The scale of this dataset is the largest human study so far in real-time human-guided RL. For each evaluator, the dataset includes 5 minutes of feedback in Bowling and 10 minutes each in Find Treasure and Hide and Seek 1v1. Each data point consists of a short trajectory comprising three consecutive image observations and the corresponding agent actions, paired with a continuous scalar

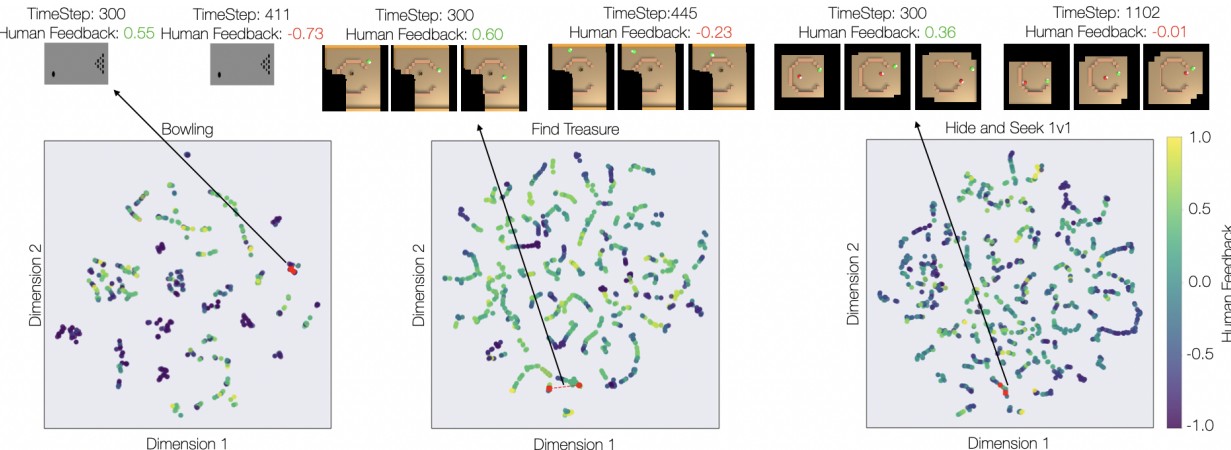

Figure 3: **Human feedback is temporally inconsistent for similar trajectories.** We visualize t-SNE embeddings of trajectory representations from evaluator 0 in all three tasks. Each point represents a trajectory, color-coded by it's corresponding scalar human feedback. Despite many trajectories being behaviorally similar (i.e., embedded closely), their feedback values vary widely. This observation highlights the temporal drift and inconsistency in human evaluations, motivating our approach to convert scalar feedback into temporally local preference pairs.

feedback signal provided by the human. To support our continual learning experiments, we also utilized training checkpoints from the start of the post-human phase, which include saved replay buffers, model weights, and optimizer states.

## 4.1 Inspiration: Human Feedback is Inconsistent Over Time

When a human evaluator observes an agent learning, we hypothesize that the criteria used to provide scalar feedback may shift over time. Early in training, humans often reward exploratory or "trying" behaviors, but later they expect more goal-directed performance. As a result, the same agent behavior might be rated positively at one point and negatively at another.

To validate our hypothesis, we processed each human evaluator's $\mathcal{D}_{\text{real-time}}$ by encoding the trajectories using a pretrained image encoder, DINO Oquab et al. (2023), followed by a t-SNE Van der Maaten & Hinton (2008) projection for visualization. The colors represent different values of the human feedback. As shown in Figure 3, there is no clear alignment between similar trajectory embeddings and scalar feedback. This trend is less obvious in Bowling due to its relatively shorter horizon and rather static observations, compared with Find Treasure and Hide and Seek 1v1. However, it's clear in the other tasks that require longer horizons and explorations, human feedback is not consistent over time. This result supports our hypothesis that feedback criteria shift as humans adjust their expectations during guidance. Visualizations from other evaluators are provided in the Appendix A.1.

This observation raises a concern for existing methods like GUIDE, which directly train a regression model to predict scalar feedback given trajectory input. If the same behavior is labeled inconsistently, regression-based models may fail to capture meaningful reward signals, resulting in unstable learning.

## 4.2 Pref-**GUIDE** `Individual`

To address the challenge of temporal inconsistency in scalar feedback, we propose PREF-GUIDE `Individual`, which converts real-time scalar feedback $\mathcal{D}_{\text{real-time}}$ into a preference-based dataset $\mathcal{D}_{\text{pref}}$.

A naive approach to handling scalar feedback would be to directly regress on it across the full training sequence. However, this fails to account for the shift in human evaluation criteria over time, and learns from

globally inconsistent and noisy signals. Our key insight is that, although feedback may drift over time, it tends to be relatively consistent within short local intervals. By focusing on temporally local comparisons and introducing a mechanism to handle ambiguity, we construct more stable and informative training signals.

We introduce two techniques to realize this idea:

**Moving Window Sampling:** We define a window

$$\mathcal{W}_i = \{(\tau_i, f_i), (\tau_{i+1}, f_{i+1}), \dots, (\tau_{i+n-1}, f_{i+n-1})\}$$

containing $n$ consecutive trajectory-feedback pairs. In practice, we set $n = 10$, which corresponds to 5 seconds of human guidance in our environments. Within this short window, we assume the human's evaluation standard is approximately stationary. We then generate $\binom{n}{2}$ trajectory pairs from each window, assigning preference labels by directly comparing their scalar feedback values.

**No Preference Range:** Even within short windows, small feedback differences may not indicate meaningful preferences. Treating any difference as a strict ordering will lead to overfitting on noisy comparisons. To address this, we introduce a no-preference threshold $\delta$, set to 5% of the total feedback range. If the difference $|f^A - f^B| < \delta$, the trajectories are treated as equally preferred and labeled with 0.5.

Using the resulting dataset $\mathcal{D}_{\text{pref}}$, we train an evaluator-specific reward model $r_\theta^{(j)}$ using the Bradley-Terry model from Section 3.2 for each evaluator $j$. This reward model is then used for continual learning once human feedback is no longer available. The full procedure is detailed in Algorithm 1.

### 4.3 Pref-**GUIDE** `Voting`

While Pref-GUIDE `Individual` mitigates temporal inconsistency within a single evaluator's feedback, it does not address the variability in feedback quality across individuals. In practice, human evaluators differ significantly in how they interpret agent behavior, the consistency of their feedback, and their attentiveness during the training process. As observed in GUIDE, some participants provide rich, informative feedback, while others are noisier, inconsistent, or even disengaged. When building real-world human-guided RL systems, relying on any single individual introduces the risk of overfitting to that evaluator's unique biases, cognitive patterns, or momentary lapses.

Our idea is to leverage the collective intelligence of multiple human evaluators to avoid the risk of overfitting. A straightforward approach might train a single reward model using pooled data from all evaluators. However, this method fails to account for evaluator-specific noise and can conflate incompatible reward signals, leading to unstable or diluted supervision.

To overcome this, we propose Pref-GUIDE `Voting`, which aggregates the predictions of independently trained evaluator-specific reward models to produce consensus preference labels. By treating each model as an independent judgment source, we can extract commonalities across diverse feedback patterns and downweight idiosyncratic or unreliable signals. Our key hypothesis is that while any individual may be noisy, the aggregated signal across multiple evaluators captures more reliable preferences, resulting in a more robust reward function for continual policy learning. The full procedure is detailed in Algorithm 2.

Specifically, instead of relying on the predicted reward from the original reward model, we query all evaluator-specific reward models $r_\theta$ $(S = 50)$ for their reward predictions on each other's input trajectories to cast a vote based on their respective judgments. These votes are averaged and then normalized to yield a consensus label in the range of $[0, 1]$, with intermediate values reflecting uncertainty or disagreement.

## 5 Experiments and Results

Our experiments aim to evaluate the effectiveness of Pref-GUIDE in improving reward model learning for continual policy training. Our experiments were performed in three challenging visual RL experiments from GUIDE Zhang et al. (2024b): Bowling, Find Treasure, and Hide and Seek 1v1. The tasks present partial observability and require long-horizon decision making, making them suitable for testing reward learning from human feedback.

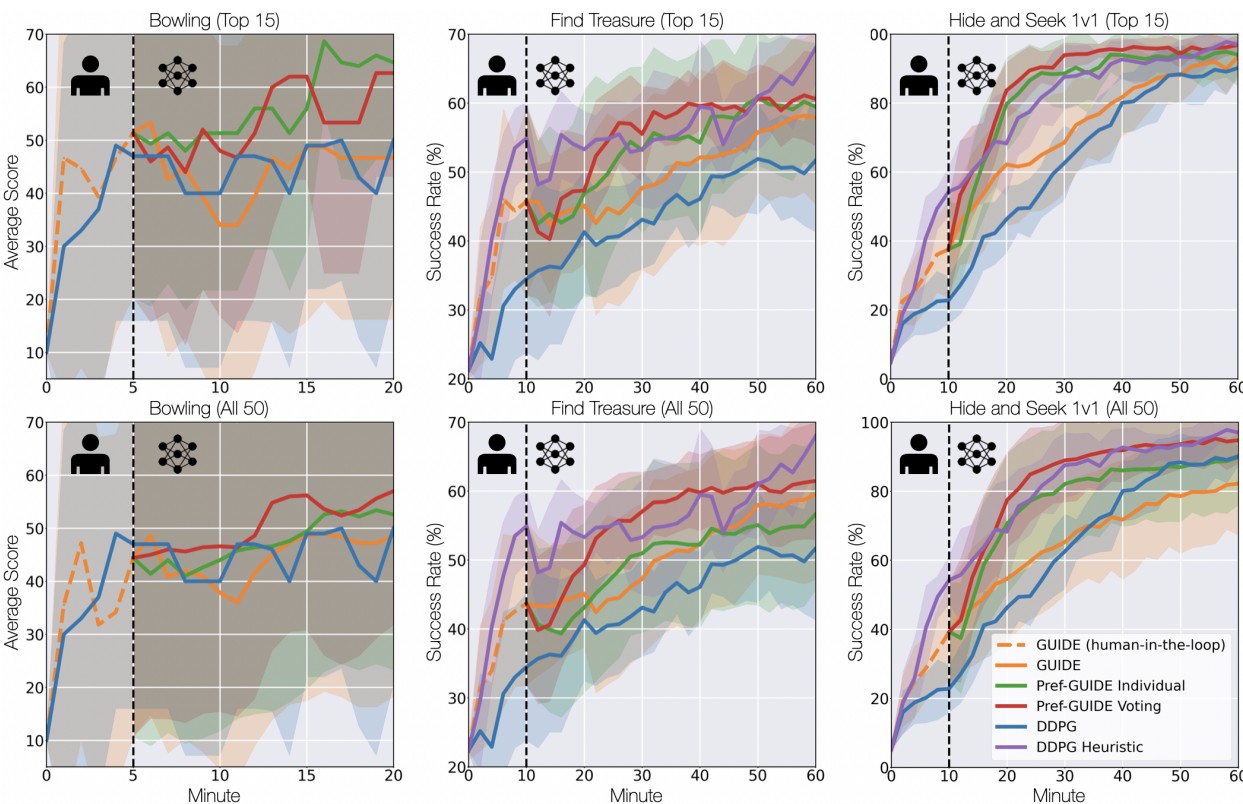

Figure 4: **Results.** Each column shows a different task. The top row reports results using only the top 15 evaluators (high-quality feedback), while the bottom row includes all 50 evaluators (mixed feedback). Curves after the dash vertical line denotes the performance during `post-human-guidance phase`. PREF-GUIDE `Individual` outperforms GUIDE when the feedback quality is high, while PREF-GUIDE `Voting` gives the best results across all conditions and even surpasses expert-designed rewards in more complex tasks.

Each run consisted of a real-time `human-in-the-loop phase`, followed by a `post-human-guidance phase` for continual learning without human input. The human-in-the-loop phase lasted 5 minutes for Bowling and 10 minutes for Find Treasure and Hide and Seek 1v1. After this, agents continued to train using learned reward models for 15 minutes for Bowling and 50 minutes for the other two tasks. To accurately track policy performance over time, we saved the policies at regular intervals for performance evaluation: every 1 minute for Bowling, and every 2 minutes for the other two tasks.

Following GUIDE, we adopt their grouping strategy in evaluations based on the cognitive test scores of human evaluators. Results are reported for two groups: the Top 15 evaluators and the All 50 evaluators. It has been found that the top 15 evaluators who obtained higher cognitive test scores provide higher quality guidance on training RL agents Zhang et al. (2024b). We conducted our ablation studies on the top 15 group to isolate design contributions under controlled experiments.

**Baselines and Evaluation Goals.** We compare against the following baselines:

- **DDPG** Lillicrap et al. (2015)**:** the base reinforcement learning algorithm used in GUIDE as the RL backbone, trained only with sparse environmental rewards.
- **DDPG Heuristic:** a version of DDPG augmented with expert-designed dense rewards, applicable only to Find Treasure and Hide and Seek 1v1. This serves as an approximate upper bound for the performance achievable with manually crafted rewards. Specifically, the dense rewards are designed based on distance measurements and exploration area tracking.

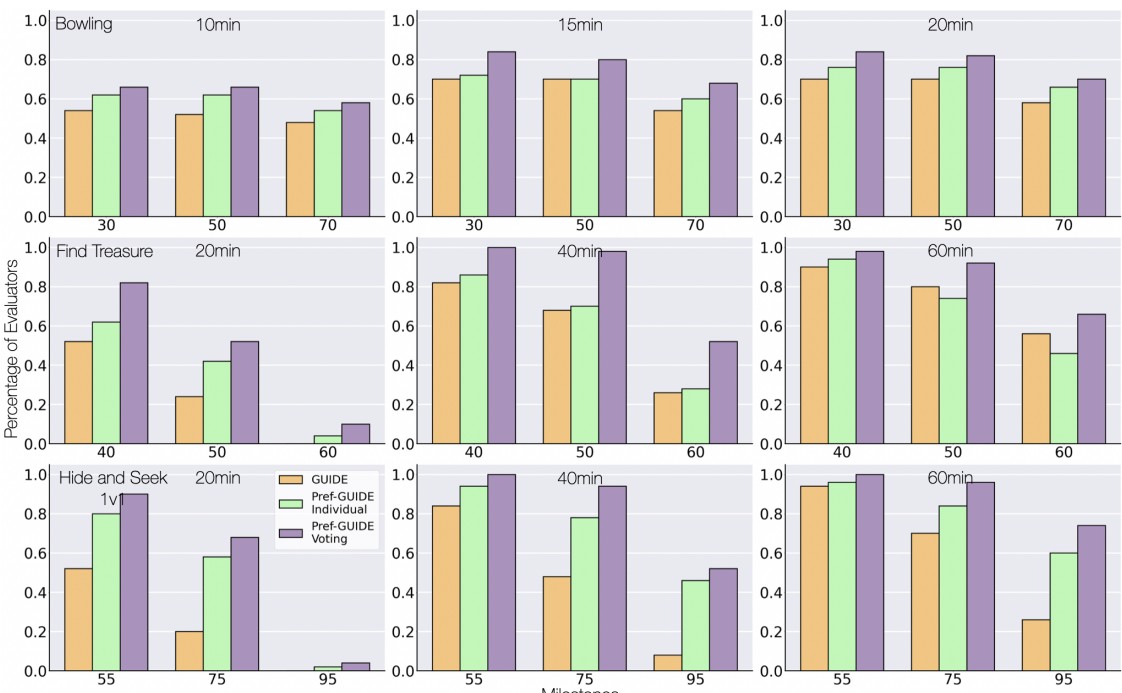

Figure 5: **Pref-GUIDE Voting enhances robustness across different evaluators.** Each subplot shows the percentage of evaluators (y-axis) whose agents reached specific performance milestones (x-axis) at different time points (title of each plot). Each row shows one task. PREF-GUIDE Voting consistently enables a larger fraction of evaluators to train high-performing agents across milestones and time.

- **GUIDE** Zhang et al. (2024b)**:** the primary baseline, which uses a regression model trained on real-time scalar feedback to guide continual learning after human supervision ends.

**Key Questions:** These baselines allow us to answer the following key questions:

- Can PREF-GUIDE outperform existing scalar-feedback regression approaches in continual training?
- How robust is it when feedback quality varies across evaluators?
- Can learned rewards from human feedback rival or surpass expert-designed dense rewards?

### 5.1   Results

Results are presented in Figure 4. When trained using feedback from the Top 15 evaluators, PREF-GUIDE Individual significantly outperforms GUIDE in all environments. This confirms our hypothesis that converting scalar feedback into localized preferences leads to more consistent and stable reward models, given that the feedback is of sufficient quality.

However, when evaluated across All 50 evaluators, PREF-GUIDE Individual shows degraded performance in the Find Treasure environment, falling slightly below GUIDE. This highlights a key limitation of relying on single-subject feedback, where biases and noise from low-quality evaluators can undermine learning. The next key question becomes how to tackle such challenges when a group of evaluators without filtering is presented.

As shown in the results, PREF-GUIDE Voting indeed can maintain high performance across both evaluator groups. By aggregating reward models across evaluators, it effectively neutralizes individual biases and noise. Importantly, PREF-GUIDE Voting not only benefits the results when all evaluators are presented without pre-filtering but also performs the best even when the Top 15 evaluators are selected with a cognitive test ranking. Furthermore, in both Find Treasure and Hide and Seek 1v1, PREF-GUIDE Voting not only

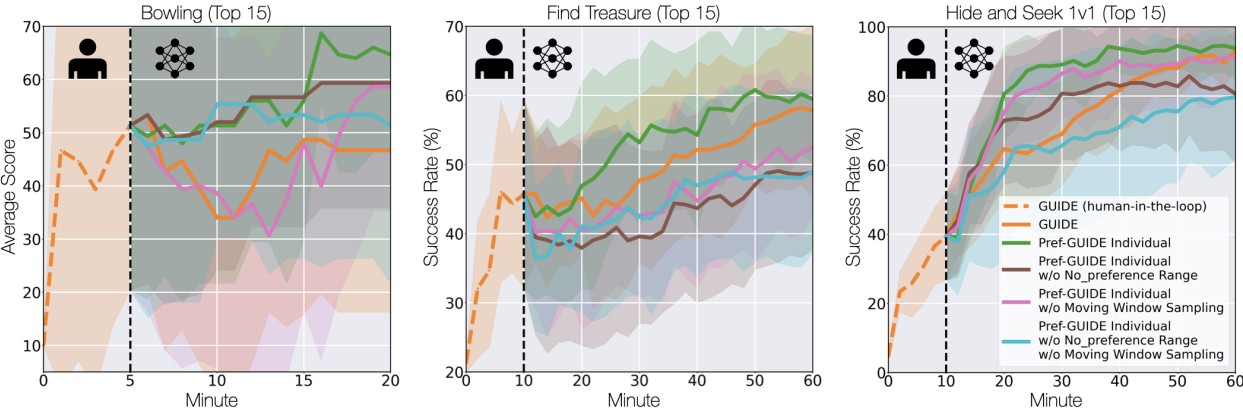

Figure 6: We evaluate the two key design choices: moving window sampling and no preference range. Removing either component leads to a noticeable drop in performance.

surpasses GUIDE but even exceeds the performance of DDPG Heuristic, suggesting that collective human feedback, when properly structured, can yield reward models more effective than expert-designed rewards.

We further compared the performance of GUIDE, PREF-GUIDE Individual (ours), and PREF-GUIDE Voting (ours) in Figure 5. For each algorithm, we computed the proportion of evaluators (y-axis) whose agents achieved a given performance threshold (x-axis). Each row shows a different task, and each column shows different training time points. PREF-GUIDE Voting consistently shows the highest performance in all settings. The results support our hypothesis that aggregating reward models at the population level yields more stable and robust supervision. This, in turn, leads to more reliable policy improvements given different evaluators.

## 5.2 Ablation Studies

**Ablation studies on Pref-GUIDE Individual.** We conducted ablations to understand the importance of the two core design choices in PREF-GUIDE Individual: (1) Moving Window Sampling, which localizes feedback comparisons, and (2) No Preference Range, which filters ambiguous pairs. As shown in Figure 6, removing either component leads to a noticeable drop in policy performance. Without the moving window, preferences are extracted from globally inconsistent feedback. The coherence is reduced. Without the no-preference margin, the model overfits to subtle, noisy changes in the feedback values, mistaking noise for meaningful preferences. These results demonstrate that both components are essential for extracting consistent and robust training signals from scalar human feedback.

**Ablation studies on Pref-GUIDE Voting.** We also evaluated the design of our consensus relabeling strategy. Our method uses normalized vote aggregation where soft preference labels reflect the degree of agreement across reward models. We first compared this to a straightforward approach, which combined all the evaluators' data to train a single reward model and used this reward model for every evaluator's continual training. Furthermore, we compared this to a simpler binary majority vote strategy, which ignores disagreement magnitude and directly applies the majority vote direction as the label.

As shown in Figure 7, normalized aggregation yields consistently better performance across all three tasks. Even though combining all the data to train a single reward model has better performance on bowling, in more complex tasks, Find Treasure and hide and Seek 1v1, it cannot outperform PREF-GUIDE Voting. Combining all the evaluators' data, although bringing more data during reward model training, also introduced conflicting labels on similar trajectories due to individual evaluator bias and noise to undermine the reward model. This suggests that capturing the confidence of the population, not just the summary of the majority, or combining all the data naively, results in smoother gradients and more informative training signals. These findings validate our design choice for robust population-level reward learning.

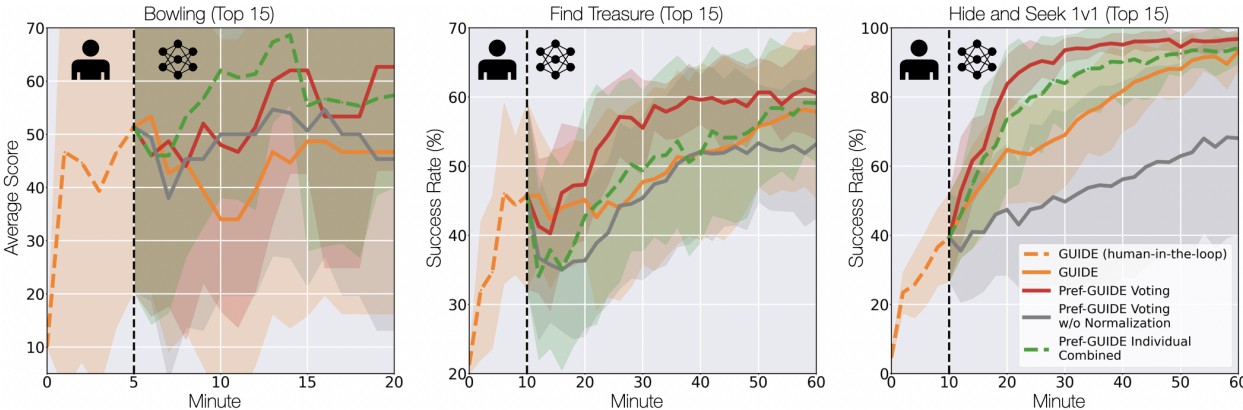

Figure 7: We compared three reward model training variants using population feedback: (1) our method with normalized voting, (2) binary majority vote relabeling, and combining all feedback data into a single training set. Normalized voting consistently outperforms others, highlighting the value of capturing the degree of agreement among evaluators rather than relying on binary or pooled labels.

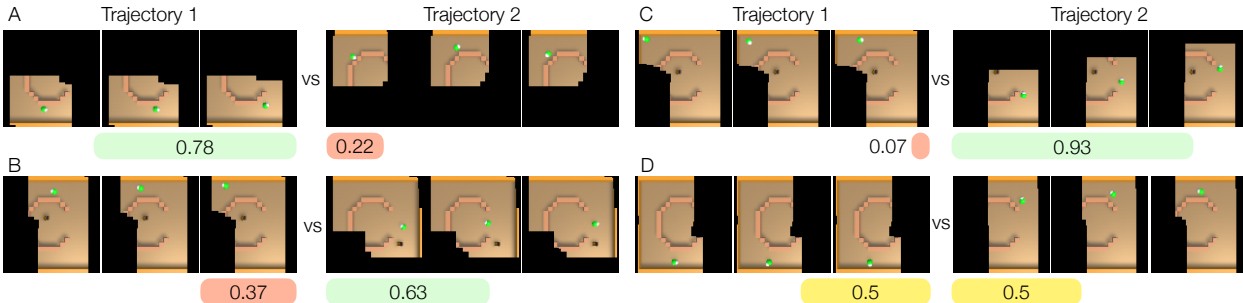

Figure 8: Examples of population voting on Find Treasure trajectories. (A–C) The population prefers exploratory behavior and movement toward discovered treasures. (D) No strong preference when both trajectories are similarly exploratory. Normalized scores reflect consensus strength.

### 5.3 Qualitative Visualizations of Voting Preferences

To better understand the behavior of PREF-GUIDE Voting, we visualize population-level preferences over several trajectory pairs from the Find Treasure environment. Figure 8 shows that population preferences align well with task-relevant behavior. For instance, in pair A, the population prefers the agent that actively explores the map over one that stalls near the starting area. In pairs B and C, the agent that moves toward the discovered treasure is preferred over one that moves away. In pair D, the population shows no preference between the two equally exploratory trajectories. These results indicate that the voting mechanism produces intuitive and consistent judgments.

### 5.4 Pref-GUIDE Voting Enables Reward Learning Agnostic to Individual Differences

To understand how different evaluator groups contribute to the final reward model in PREF-GUIDE Voting, we analyzed the voting contributions of subgroups categorized by cognitive test scores: Top 15 (highest performers), Middle 20, and Bottom 15. Table 5.4 shows the proportion of each group's contributions to winning votes across three environments.

Interestingly, we find that each subgroup's contribution is roughly proportional to its representation in the overall population. This suggests that no subgroup disproportionately dominates the voting outcomes and that the final preference labels reflect a balanced aggregation across all evaluators.

| Group | Bowling | Find Treasure | Hide and Seek 1v1 |
|---|---|---|---|
| Top 15 (30%) | 30.78% | 29.85% | 30.96% |
| Middle 20 (40%) | 39.22% | 40.38% | 40.08% |
| Bottom 15 (30%) | 30.00% | 29.77% | 28.96% |

Table 1: Distribution of winning votes across evaluator groups. Each group's contribution to the consensus aligns with its population size, indicating balanced influence across cognitive subgroups.

However, this observation raises a deeper question. When comparing the performance of PREF-GUIDE `Voting` between Top 15 and All 50 evaluators in Figure 4, they are quite similar. This is not the case for other methods and baselines, where Top 15 evaluators always generate higher performance on the guided agents than All 50 evaluators. The question is: if the lower-performing groups in cognitive tests contribute equally to the voting outcomes, why does PREF-GUIDE `Voting` under All 15 evaluators still match the performance of Top 15 feedback alone?

We believe that the answer lies in the balancing effect of the consensus-based aggregation. While the Top 15 group alone yields stronger raw feedback signals, as reflected in the superior performance of PREF-GUIDE `Individual` on this subgroup, the PREF-GUIDE `Voting` mechanism can filter and dilute the noise introduced by less consistent evaluators, while still preserving useful signals from across the population. This trade-off results in a robust reward model that performs similarly whether trained on just the Top 15 or on all 50 evaluators.

Our results suggest that PREF-GUIDE `Voting` makes the agent's performance **more agnostic to the quality of individual human feedback**. It balances signal and noise in a way that enables learning from a broad population, without being overly sensitive to individual differences among human evaluators.

## 6 Conclusion

We introduced PREF-GUIDE, a method that converts real-time scalar human feedback into preference-based data to train reward models for continual policy learning. PREF-GUIDE `Individual` mitigates temporal inconsistency by transforming continuous feedback labels into moving-window sampled preference labels. PREF-GUIDE `Voting` enhances robustness by aggregating reward models across individuals into a consensus-driven signal. Across three visual RL environments, PREF-GUIDE `Individual` outperforms regression-based baselines such as GUIDE when feedback is of high quality, and PREF-GUIDE `Voting` maintains strong performance even under noisy conditions, surpassing both GUIDE and expert-designed dense rewards. Together, these results highlight the scalability and stability of structured preference learning from real-time human input.

As for the limitation of our method, for the two design choices of PREF-GUIDE `Individual`: no-preference range and moving window, we assumed that the same set of parameters is sufficient to model different human evaluators. Though our results showed that the same set of parameters performed well, these parameter selections may be further improved. For instance, more optimal subject-specific no-preference ranges and moving window sizes could potentially be inferred through a warm-up phase of human guidance. Furthermore, for every human evaluator, we trained a separate model. This model is rather small, and the additional training overhead is minimal. However, it is possible that such a model can add more training overhead when the reward model size needs to be large. Exploring more efficient reward models can be an interesting future direction. For PREF-GUIDE `Voting`, more algorithm improvements or more sophisticated aggregation methods can also be studied to incorporate the human guidance from all subjects.

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

# A  Appendix

## A.1  Visualization of Trajectories vs Human Feedback

We provided the visualization of trajectories vs human feedback for all evaluators in Figure 9. For each evaluator's plot (evaluator's id is on the top left corner for each set of plots), the tasks are Bowling, Find Treasure, and Hide and Seek 1v1 (from left to right).

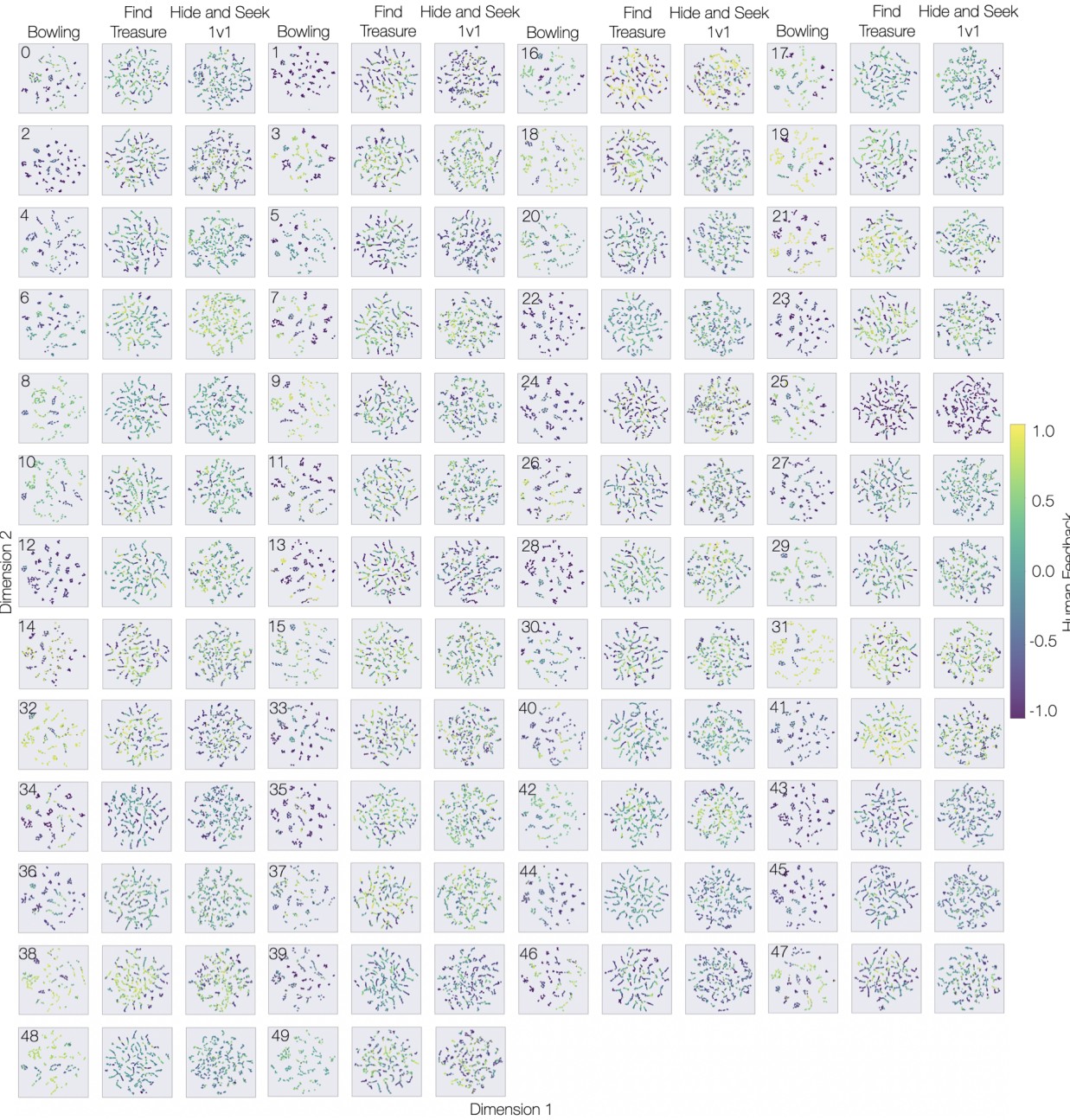

Figure 9: Visualization of Trajectories vs Human Feedback for All 50 Evaluators. There is no clear alignment for most evaluators in the three tasks.

Furthermore, we also visualize the trajectories that correspond to the neighbor points in the embedding space to further verify our hypothesis. In Figure 10, 11, and 12, for each task, we uniformly selected 8 anchor trajectories, and found the closest trajectory to every one of them in t-SNE space that is at least 100 timesteps apart to avoid temporal similarity. We visualized these trajectories along with the corresponding human feedback. As shown in the resulting trajectories, despite that they are semantically and visually very similar or near identical within the same trajectory, the human feedback is very distinct at different timesteps. These observations show strong evidence that human feedback is not consistent.

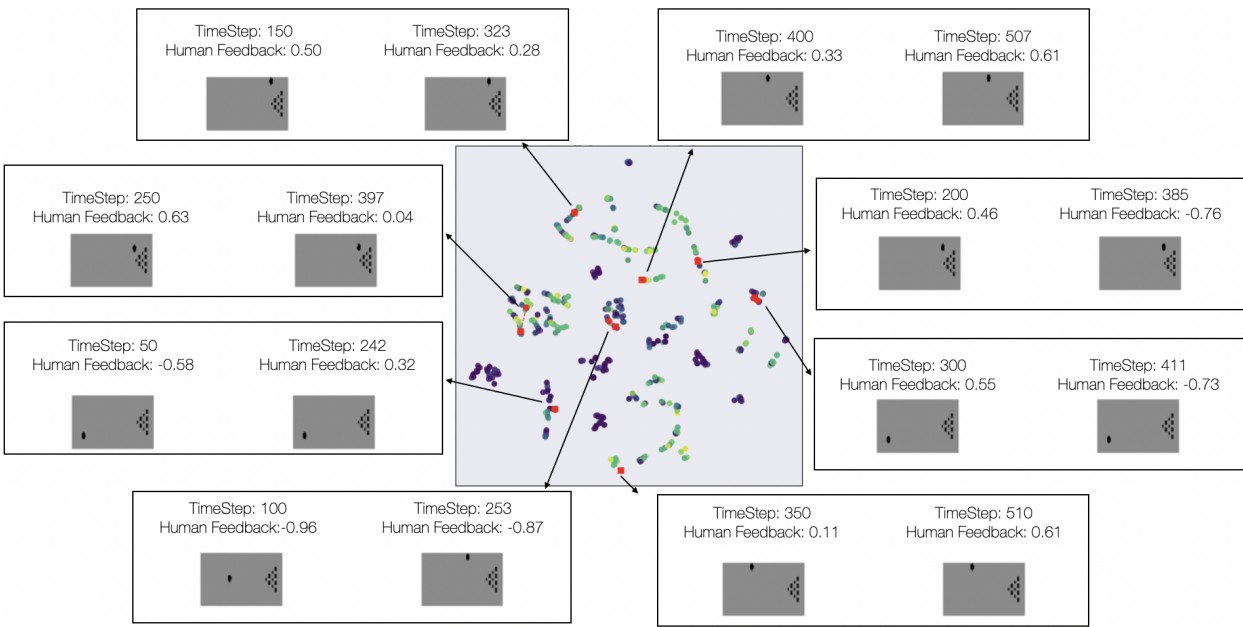

Figure 10: Visualization of Bowling from Subject 0. The trajectories that are semantically or visually similar to each other at different timesteps receive very distinct human feedback values.

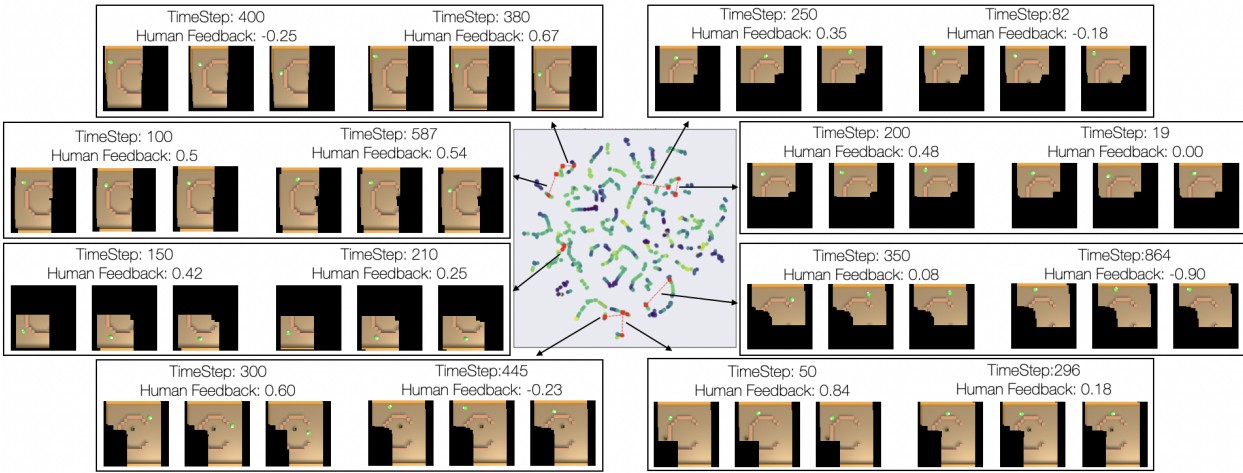

Figure 11: Visualization of Find Treasure from Subject 0. The trajectories that are semantically or visually similar to each other at different timesteps receive very distinct human feedback values.

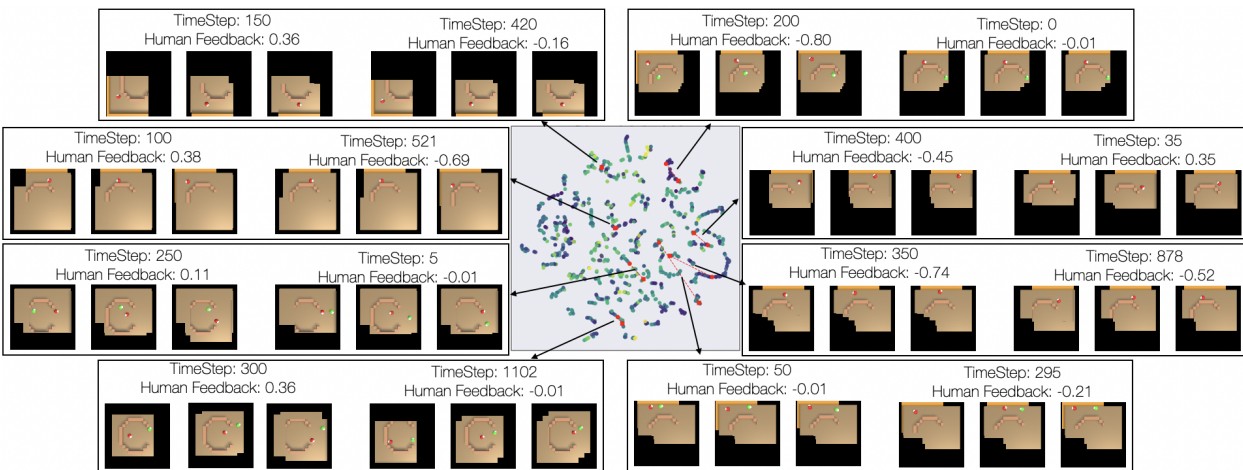

Figure 12: Visualization of Hide and Seek 1v1 from Subject 0. The trajectories that are semantically or visually similar to each other at different timesteps receive very distinct human feedback values.

## A.2 Pref-**GUIDE** `Individual` **Algorithm**

The algorithm of PREF-GUIDE `Individual` is summarized in Algorithm 1.

---

**Algorithm 1** PREF-GUIDE `Individual`

---

**Require:** Real-time dataset $\mathcal{D}_{\text{real-time}} = \{(\tau_i, f_i)\}_{i=1}^N$, window size $n$, no preference range $\delta$
1: Initialize $\mathcal{D}_{\text{pref}} \leftarrow \emptyset$, $r_\theta$
2: **for** $i = 1$ to $N - n + 1$ **do**
3:     Define window $\mathcal{W}_i = \{(\tau_j, f_j)\}_{j=i}^{i+n-1}$
4:     **for** each pair $(\tau^A, f^A), (\tau^B, f^B) \in \mathcal{W}_i$ such that $A < B$ **do**
5:         **if** $|f^A - f^B| < \delta$ **then**
6:             $\mathcal{D}_{\text{pref}} \leftarrow \mathcal{D}_{\text{pref}} \cup \{((\tau^A, \tau^B), 0.5)\}$
7:         **else if** $f^A > f^B$ **then**
8:             $\mathcal{D}_{\text{pref}} \leftarrow \mathcal{D}_{\text{pref}} \cup \{((\tau^A, \tau^B), 1)\}$
9:         **else**
10:            $\mathcal{D}_{\text{pref}} \leftarrow \mathcal{D}_{\text{pref}} \cup \{((\tau^A, \tau^B), 0)\}$
11:         **end if**
12:     **end for**
13: **end for**
14: Train $r_\theta$ using $\mathcal{D}_{\text{pref}}$
15: Use $r_\theta$ for continual learning

---

## A.3 Pref-**GUIDE** `Voting` **Algorithm**

The algorithm of PREF-GUIDE `Voting` is summarized in Algorithm 2.

---

**Algorithm 2** PREF-GUIDE Voting

---

**Require:** Real-time dataset $\mathcal{D}_{\text{real-time}} = \{(\tau_i, f_i)\}_{i=1}^N$, window size $n$, no preference range $\delta$, evaluator-specific PREF-GUIDE `Individual` reward models $\{r_\theta^{(j)}\}_{j=1}^S$

1: Initialize $\mathcal{D}_{\text{pref-voting}} \leftarrow \emptyset$
2: **for** $i = 1$ to $N - n + 1$ **do**
3:     Define window $\mathcal{W}_i = \{(\tau_j, f_j)\}_{m=i}^{i+n-1}$
4:     **for** each pair $(\tau^A, f^A), (\tau^B, f^B) \in \mathcal{W}_i$ such that $A < B$ **do**
5:         Initialize vote count $c \leftarrow 0$
6:         **for** $j = 1$ to $S$ **do**
7:             **if** $|r_\theta^{(j)}(\tau^A) - r_\theta^{(j)}(\tau^B)| < \delta$ **then**
8:                 $c \leftarrow c + 0.5$
9:             **else if** $r_\theta^{(j)}(\tau^A) > r_\theta^{(j)}(\tau^B)$ **then**
10:                 $c \leftarrow c + 1$
11:             **else**
12:                 $c \leftarrow c + 0$
13:             **end if**
14:         **end for**
            $\mathcal{D}_{\text{pref-voting}} \cup \{((\tau^A, \tau^B), \frac{c}{S})\}$
15:     **end for**
16: **end for**
17: Train final voting-based reward model $R_\theta$ using $\mathcal{D}_{\text{pref-voting}}$
18: Use $R_\theta$ for continual learning

---

### A.4 Experiment Details

Each experimental result was averaged over 5 seeds. The dense rewards for Hide and Seek 1v1 and Find Treasure were defined based on the visibility of the treasure or hider and the distance between the agent and the target.

### A.5 Hardware Requirement

We conducted the experiments using NVIDIA A100 and NVIDIA RTX A6000. However, our experiment can be run on a single desktop with a single GPU, such as NVIDIA RTX 4070.

