# OpenReview forum: "Pref-GUIDE: Continual Policy Learning from Real-Time Human Feedback via Preference-Based Learning"
_TMLR — Accepted by TMLR_

### Review · Reviewer_gwVU · 2025-08-04

**Summary Of Contributions:**

Consider the problem of real-time human-guided reinforcement learning. One algorithm that exists for this problem is called GUIDE [Zhang et al. (2024b)], and it consists of two stages:

1. **Human-in-the-loop phase-** the agent learns from a combination of sparse terminal rewards and the real-time scores in [-1,1] from human evaluators.

2. **Post-human-guidance phase-** after feedback stops, GUIDE trains a reward model on the collected data and uses its predictions as a dense reward signal to continue policy learning without further human input.

This paper improves the post-human-guidance phase of GUIDE by addressing two inherent problems: (a) inconsistent feedback from the same human evaluator over time, and (b) variability across evaluators (e.g., due to cognitive ability).

**Contributions**

- **Pref-GUIDE individual-** converts scalar feedback into preference-based comparisons while mitigating problem (a) using a moving-window sampling strategy under the assumption of approximate stationarity within each window.

- **Pref-GUIDE Voting-** to address problem (b), it trains a separate model per evaluator and aggregates their predictions into a consensus reward model.

- The authors test these methods on multiple RL environments: Bowling, Find Treasure, and Hide and Seek 1v1, and show that they improve performance compared to GUIDE significantly.
- The paper demonstrates competitive performance with DDPG/DDPG heuristic, with a significant improvement in Bowling.


**Strengths**

- Pref-GUIDE Individual and Pref-GUIDE Voting provide solutions for temporal inconsistency and evaluator bias, and I’m unaware of other existing solutions for PbRL.

- The paper is well-written.

**Weaknesses**

- The solutions suggested are relatively trivial.

-  The only significant improvement over the DDPG + DDPG heuristic appears in a single environment (Bowling) out of the three tested.

- In Pref-GUIDE Individual, pairs of scalar rewards in [-1,1] are translated to {-1,0,1}, where 0 indicates no preference. Two rewards are mapped into 0 iff their difference is at most 5%. The choice of a 5% threshold for mapping small reward differences to “no preference” seems arbitrary. In particular, the paper lacks justification and does not explore statistically grounded thresholds (e.g., based on standard deviation or percentiles).

- [Minor] Pref-GUIDE Individual may only correct for linear drift. Nonlinear temporal bias could still degrade performance.


- See more weaknesses in `Requested Changes’.

**Additional Comments:**

T-SNE is popular in vision and high-dimensional numeric tasks, but can you provide evidence that it meaningfully captures similarity among Bowling trajectories? In particular, have you verified, quantitatively or via human judgment, that trajectories close in t-SNE space receive similar preference scores? Is it possible that some trajectories appear adjacent in the embedding but actually should have very different scores?
To support your claim of inconsistent feedback over time (Section 4.1), do you have score data from the same evaluator on identical trajectories at multiple time points?

**Audience:**

Yes

**Audience Explanation:**

I think PbRL is a topic of interest to the TMLR’s audience, and this paper might improve the state of the art in a relevant task and improves an existing algorithm in 3 tasks.

**Broader Impact Concerns:**

There are no ethical implications of the work that I can think of.

**Claims And Evidence:**

No

**Claims Explanation:**

In Section 4.1, the authors hypothesize that a trajectory rated positively at one point might be rated negatively later, but the only supporting evidence (Figures 3 and 9) do not track individual ratings as a function of time. Without any per-trajectory time-series analysis, such as plots of the same trajectory’s scores across sessions or statistical tests for rating drift, the hypothesis remains unsubstantiated. The authors should include direct temporal evidence to convincingly support their claim.

**Requested Changes:**

**Critical for acceptance**

*Major*

- In Section 4.1, the authors hypothesize that the same agent might rate the same trajectory positively in the beginning and negatively at a later stage. The only supporting evidence (Figures 3 and 9) does not account for temporal variation.  I think the authors should include per-trajectory rating time series or time-point analysis to validate this claim.

- I think the authors should include the DDPG + DDPG heuristic baseline in Figures 5-7 for fair comparison.

*Minor*

- Figure 3 (and Figure 9 in the appendix) shows Bowling receiving relatively consistent feedback despite the claim of inconsistency across all tasks. I think the authors should add a discussion explaining this apparent contradiction.

- Figure 4 shows significant improvement only for Bowling relative to DDPG/DDPG heuristic, yet Figure 3 suggests Bowling’s feedback is relatively consistent. I think the authors should reconcile these observations in the discussion.

- I think the authors should add a discussion of computational overhead and scalability when training separate models per evaluator (I would assume it is not very significant, but it’s still missing).

- I think the authors should add a discussion on how the window size was chosen.

**Suggestions to strengthen the work**

- I’m not surprised that Pref-GUIDE Voting yields a more stable and robust model. I think using smarter aggregation techniques such as boosting would be more beneficial.

---

> ### Author Response · Authors · 2025-08-07
> **Response to Reviewer gwVU (part 1)**
>
> We thank the reviewer for the thoughtful comments. We would like to address all of your concerns and questions below with point responses.
> (part1)
> >**“The solutions suggested are relatively trivial.”**
> >
>
> We respectfully disagree with this comment. Our method is conceptually intuitive but highly effective as evidenced by our experiments. We believe that this is in fact a merit of our approach instead of a limitation. Moreover, to the best of our knowledge, we are the first to propose converting real-time human feedback into preference pairs and formulating continual learning of real-time human-guided RL as preference-based learning. Furthermore, although many previous studies[1,2,3] involve multiple evaluators labeling data, none have considered population-level voting as a form of aggregation. These two novel components brought key advantages to our methods with careful ablation studies to demonstrate them. Both our method and experiments suggest that our contribution and solutions are not trivial.
>
> >**“The only significant improvement over the DDPG + DDPG heuristic appears in a single environment (Bowling) out of the three tested.”**
> >
> We would like to clarify that the reviewer may have misread Figure 4. **Our method outperforms all three tests, as shown in Figure 4**.
>
> As shown in GUIDE [4], the authors used DDPG as the backbone algorithm due to their stronger performance on visual-based RL. This means that the algorithm improvements of GUIDE were on top of DDPG. DDPG Heuristic refers to DDPG augmented with an expert-designed reward function, which serves as an upper bound on performance by incorporating human expert knowledge, something generally unavailable for many tasks. We followed this in our paper. And our goal is for Pref-GUIDE to outperform both DDPG and GUIDE.
>
> We highlight that Pref-GUIDE Individual outperforms both GUIDE and DDPG across all three tasks, as evidenced by the plots of the top 15 evaluators in the first row of Figure 4. We further observe that Pref-GUIDE Individual suffers from variability in feedback quality, which motivated us to propose Pref-GUIDE Voting as a solution. Pref-GUIDE Voting consistently outperforms GUIDE and DDPG across both the top 15 evaluators and all 50 evaluators, as shown throughout Figure 4.
>
> Additionally, we show that Pref-GUIDE Voting can even outperform the hand-designed reward **in complex tasks, including Find Treasure and Hide and Seek 1v1**, as demonstrated in the last two columns of Figure 4.
>
> >**“In Pref-GUIDE Individual, pairs of scalar rewards in [-1,1] are translated to {-1,0,1}, where 0 indicates no preference. Two rewards are mapped into 0 iff their difference is at most 5%. The choice of a 5% threshold for mapping small reward differences to “no preference” seems arbitrary. In particular, the paper lacks justification and does not explore statistically grounded thresholds (e.g., based on standard deviation or percentiles).”**
> >
> The 5% threshold is a design choice intended to reflect the idea that small variations in real-time scalar feedback are often not meaningfully distinguishable by humans. This number reflects the ambiguity in human preference labels. We would like to clarify that this is a common practice in PbRL with many past papers selecting 5-15% as a hyperparameter [1,2,3]. We hence follow the existing practice. Our algorithm outperforms all baselines in all three tasks with the same hyperparameters, suggesting that our method is not very sensitive to this hyperparameter.. In our ablation study on Pref-GUIDE Individual (Section 5.2 and Figure 6), we also demonstrated that removing this no-preference range degrades the performance of Pref-GUIDE Individual. We will highlight this as a hyperparameter in our paper.

---

> ### Author Response · Authors · 2025-08-07
> **Response to Reviewer gwVU (part 2)**
>
> (part2)
>
> >**“Pref-GUIDE Individual may only correct for linear drift. Nonlinear temporal bias could still degrade performance.”**
> >
>
> We are not sure if the reviewer is referring to our linear moving window. Though our method moves the window linearly in time, the preference data conversion only happens within each window. Modeling human drift is still an open question, and we leave the exploration of more complex methods in the future. Our method is the first demonstration of handling such issues in real-time human-guided RL and outperforms all other baselines in all our tasks.
>
> >**“In Section 4.1, the authors hypothesize that a trajectory rated positively at one point might be rated negatively later, but the only supporting evidence (Figures 3 and 9) do not track individual ratings as a function of time. Without any per-trajectory time-series analysis, such as plots of the same trajectory’s scores across sessions or statistical tests for rating drift, the hypothesis remains unsubstantiated. The authors should include direct temporal evidence to convincingly support their claim.”**
> >
>
> >**“In Section 4.1, the authors hypothesize that the same agent might rate the same trajectory positively in the beginning and negatively at a later stage. The only supporting evidence (Figures 3 and 9) does not account for temporal variation. I think the authors should include per-trajectory rating time series or time-point analysis to validate this claim.”**
> >
>
> >**“T-SNE is popular in vision and high-dimensional numeric tasks, but can you provide evidence that it meaningfully captures similarity among Bowling trajectories? In particular, have you verified, quantitatively or via human judgment, that trajectories close in t-SNE space receive similar preference scores? Is it possible that some trajectories appear adjacent in the embedding but actually should have very different scores? To support your claim of inconsistent feedback over time (Section 4.1), do you have score data from the same evaluator on identical trajectories at multiple time points?”**
> >
>
>
> We thank the reviewer for the thoughtful suggestions. All three comments raise a similar concern, so we address them together. We first would like to clarify our methods of plotting the trajectory embedding vs human feedback.  To obtain the t-SNE embeddings, we first encode trajectories using a strong pretrained vision model, DINO, which is well known to preserve semantic features from their encoded features in images. This ensures that trajectories with visually similar structure are embedded closely in the latent space. While t-SNE is not without limitations, it is widely used in the field to reduce the dimension of the embeddings for visualization.
> To further address your concerns, we are happy to report that we conducted further per-trajectory analysis and show that similar trajectories received different scores at different time steps. We provided an anonymous link, https://github.com/kdsfjhkadshkja/Pref-GUIDE-Rebuttal, where we included a modified version of Figure 3 (Modified Figure 3.png) and 3 visualizations of our newly conducted per-trajectory analysis.
>
> To obtain these analyses, we took a quantitative and systematic method instead of a qualitative method. Specifically, in the 3 visualizations (Bowling.png, Find Treasure.png, and Hide and Seek 1v1.png), for each task, we randomly selected 8 anchor trajectories where each trajectory include sub-trajectories at timesteps 50, 100, …, and 400, and found the closest trajectory to every one of them in t-SNE space that are at least 100 timesteps apart to avoid temporal similarity. We visualized these trajectories along with the corresponding human feedback. As shown in the resulting trajectories, despite that they are semantically and visually very similar or near identical within the same trajectory, the human feedback is very distinct at different timesteps. The only differentiating factor here is their timesteps.  We hope this addresses your concerns about whether data points close in the t-SNE embedding are actually similar, and whether human feedback varies across them due to time differences. We will include these three plots in our Appendix and included a subset of them into our updated Figure 3 due to space considerations with a link to the comprehensive plots in Appendix
>
> Having the subjects see the exact same trajectory is not feasible in our setting. Since our RL policy is updated and trained over time, the trajectories by rolling out the policy are naturally different. Enforcing such a constraint is not feasible and does not follow our problem setting.

---

> ### Author Response · Authors · 2025-08-07
> **Response to Reviewer gwVU (part 3)**
>
> (part3)
> >**“I think the authors should include the DDPG + DDPG heuristic baseline in Figures 5-7 for fair comparison.”**
> >
> Thanks for your suggestion. Figure 6-7 shows ablation studies to show the effectiveness of our proposed algorithm improvements, where our method’s results are the same in Figure 4. We have compared with DDPG and DDPG Heuristic in Figure 4, where our method outperforms both of them across all three tasks. We will include DDPG and DDPG Heuristic in Figure 5.
>
> >**“Figure 3 (and Figure 9 in the appendix) shows Bowling receiving relatively consistent feedback despite the claim of inconsistency across all tasks. I think the authors should add a discussion explaining this apparent contradiction.”**
> >
> Thanks for your suggestion. We will add a sentence in the appendix to discuss the relative consistency in bowling.
> We agree with this observation since Bowling involves less temporal exploration and shorter horizons than the other tasks. We will add one sentence to indicate this as follows:“ This trend is less obvious in Bowling due to its relatively shorter horizon and rather static observations, compared with Find Treasure and Hide and Seek v1. However, it’s clear in the other tasks that require longer horizons and explorations, human feedback is not consistent over time. ”
>
> >**“Figure 4 shows significant improvement only for Bowling relative to DDPG/DDPG heuristic, yet Figure 3 suggests Bowling’s feedback is relatively consistent. I think the authors should reconcile these observations in the discussion.”**
> >
>
> As discussed above, Figure 4 demonstrates that **Pref-GUIDE consistently outperforms both DDPG and GUIDE across all three tasks**, not just Bowling.
>
> >**“I think the authors should add a discussion of computational overhead and scalability when training separate models per evaluator (I would assume it is not very significant, but it’s still missing).”**
> >
>
> Thanks for your suggestions. We will add this sentence in the conclusion section as a discussion of the computational overhead and scalability.
> “Furthermore, in this work, for every human evaluator, we trained a separate model. This model is rather small and the additional training overhead is minimum. However, it is possible that such a model can add more training overhead when the reward model size needs to be large. Exploring more efficient reward models can be an interesting future direction.”
>
> >**“I think the authors should add a discussion on how the window size was chosen”.**
> >
> The window size was set to 10 frames (5 seconds) as a design choice. This is a hyperparameter. Our ablation studies show that not using this window but relying on the full sequence harms the performance. Modeling human consistency is still an open question as this can be different in different tasks. We use the same hyperparameter in our study across all three tasks while showing superior performance, showing that our algorithm is not very sensitive to this parameter. We will clarify this in the paper.
>
>
> >**“I’m not surprised that Pref-GUIDE Voting yields a more stable and robust model. I think using smarter aggregation techniques such as boosting would be more beneficial.”**
> >
>
> We appreciate the reviewer’s thoughtful feedback. We chose to use a simple voting scheme rather than boosting or more complex ensemble methods to maintain an effective yet simple aggregation strategy. We would also like to clarify that our aggregation is different from the traditional ensemble learning techniques. The goal of this aggregation is not to directly produce better reward predictions through an **ensembled model**, but rather to generate softer, consensus-based labels that can be used to relabel the dataset for training **improved individual reward models**. Voting across the population offers a direct and robust way to achieve this. Additionally, boosting requires evaluating the accuracy of individual weak learners on the same dataset to change their weights, which is not straightforward in our setting: each model is trained on feedback from a different human subject with unique preferences, making validation loss an unfair metric for comparison. That said, we agree this is an important direction for future work, and to address your concern, we will add the following sentence to the conclusion:
> “A more sophisticated strategy of assembling models can be studied to better aggregate population power. ”

---

> ### Author Response · Authors · 2025-08-07
> **Response to Reviewer gwVU (part 4)**
>
> (part4)
>
> *References*
>
> *[1] Paul F Christiano, Jan Leike, Tom Brown, Miljan Martic, Shane Legg, and Dario Amodei. Deep reinforcement learning from human preferences. Advances in neural information processing systems, 30, 2017.*
>
> *[2] Borja Ibarz, Jan Leike, Tobias Pohlen, Geoffrey Irving, Shane Legg, and Dario Amodei. Reward learning from human preferences and demonstrations in atari. Advances in neural information processing systems, 31, 2018.*
>
> *[3] Kimin Lee, Laura Smith, and Pieter Abbeel. Pebble: Feedback-efficient interactive reinforcement learning via relabeling experience and unsupervised pre-training. arXiv preprint arXiv:2106.05091, 2021a.*
>
> *[4] Lingyu Zhang, Zhengran Ji, Nicholas R Waytowich, and Boyuan Chen. Guide: Real-time humanshaped agents. In A. Globerson, L. Mackey, D. Belgrave, A. Fan, U. Paquet, J. Tomczak, and C. Zhang (eds.), Advances in Neural Information Processing Systems, volume 37, pp. 138959–138980. Curran Associates, Inc., 2024b. URL https://proceedings.neurips.cc/paper_files/paper/2024/ file/facf3192e99ce60c0ef5ed4067b72f68-Paper-Conference.pdf.*

---

> > ### Comment · Reviewer_gwVU · 2025-09-19
> >
> > Thank you for your thorough and explanations. You have satisfactorily addressed all of my concerns.

---

> ### Author Response · Authors · 2025-09-20
> **Thank you for your response**
>
> We are glad we addressed all your questions and concerns. Thank you!

---

### Review · Reviewer_Pf2B · 2025-08-09

**Summary Of Contributions:**

This paper introduces pref-GUIDE, a method for addressing human evaluator inconsistency and individual bias during real time feedback in preference based RL. This method builds on GUIDE and focus on the "continual" learning when the human evaluator is no longer available and only the dataset from the previous human-in-the-loop phase is available. The proposed method for addressing temporal inconsistency is by splitting human feedback into temporally adjacent windows and generating preference pairs from the windows. The proposed method for addressing individual bias is by training a reward model on each evaluator's data and aggregating the reward scores during continual policy training. Experiments are conducted on partially observable visual RL environments. The results demonstrate the advantage of the proposed methods over GUIDE and DDPG trained on true sparse environment rewards and expert designed heuristic rewards. Ablations demonstrate the necessity of the proposed improvements over GUIDE.

**Audience:**

Yes

**Audience Explanation:**

This paper proses an improvement over an existing algorithm for real-time learning from human feedback, which is a topic of general interest. The focus on adapting to human bias and inconsistency is a popular topic within this community.

**Claims And Evidence:**

Yes

**Claims Explanation:**

The experiments show clear advantage of the proposed methods (e.g., figure 5). The ablations further demonstrate the contribution of each proposed components.

**Requested Changes:**

* What's "real-time" about this feedback paradigm? Is it because human evaluators need to give a feedback $f$ at every time step? Wouldn't this be too slow to be practical, since the frame rate of most tasks is too high for humans to make a judgement?
* What exactly is the form of the reward model? Is it $R(\tau) = \sum_{t}R(s_{t}, a_{t})$?
* Why would the moving window sampling help mitigate temporal inconsistency? I would imagine training a separate BT model on each temporal window would address inconsistency, at the expense of have potentially too many models. However, if you only use the windows to sample preference pairs, and train a BT model under the assumption that the reward model underlying all comparisons is the same, how can this address inconsistency?
* For the ablation results in figure 6, is voting applied to generate the figure? Or is the data generated by different evaluators considered as a single evaluator for training the reward model here? I think the authors should clarify this. It seems to me that voting has the highest contribution to performance. If not the former, I think the authors should do an ablation on pref-GUIDE individual + voting with no temporal sampling and no preference range.
* How exactly are the rewards aggregated for training the policy?

Minor
* How many seeds did you use to repeat each experiment?
* What RL algorithm did you use for the proposed method?

---

> ### Author Response · Authors · 2025-08-13
> **Response to Reviewer Pf2B (part 1)**
>
> We thank the reviewer for your thoughtful comments. We would like to address all of your concerns and questions below with point responses.
>
> >**What's "real-time" about this feedback paradigm? Is it because human evaluators need to give a feedback  at every time step? Wouldn't this be too slow to be practical, since the frame rate of most tasks is too high for humans to make a judgement?**
>
> Yes. In our setup, human evaluators provide feedback while observing the agent’s behavior in real time, adapted from prior human-in-the-loop RL paradigms [1–5]. “Real-time” here means that feedback is given as the environment unfolds at wall-clock speed, not necessarily at the simulator’s maximum frame rate. This is a realistic computational framework in real-time human-guided RL for several reasons:
>
> - Feasibility for humans. The environment is rendered at a human-perceivable rate (e.g., video game speed), allowing evaluators to make judgments without needing to react to every single frame. As shown in GUIDE [5], humans (N=50 subjects) can reliably provide useful feedback under these rates, even for complex visual RL tasks. Therefore, it has been shown that humans can provide effective feedback in multiple prior works [1-5].
> - The online policy rollouts presented to humans are run at normal speed for evaluation purposes, but policy training can still occur in parallel and at higher simulator speeds once human guidance ends. Our method uses a post-human-guidance phase where training proceeds as in standard RL without human latency constraints.
> - Following prior work [1–5], we associated each scalar feedback signal with a short sequence of observation–action pairs rather than a single frame. These algorithms, including ours, enable faster and better convergence without precise single-timestamp association.
> - Most RL environments provide a human “viewing” interface for humans to inspect or evaluate the policy behaviors. This is also commonly used by experts in reward engineering. Moreover, all our tasks are visual RL tasks that take in visual observations, which makes it more reasonable to have a human visualization interface that runs at a frequency that humans can provide effective judgments. Therefore, our problem formulation does not alter or require special assumptions compared with established work.
>
>
> >**What exactly is the form of the reward model? Is it  $R(\tau) = \sum_{t} R(s_t, a_t)$**
>
> No. In our setting, the reward model takes an entire trajectory as input and outputs a single reward for the whole trajectory. Instead of predicting a reward for each frame and summing them to obtain the trajectory reward, we train the model directly to predict the total reward. We chose this approach to match the setting in GUIDE [5], where, during the post-human-guidance phase, the reward model takes three consecutive frames as input and predicts a reward. To ensure a fair comparison, we designed our reward model to have the same structure as GUIDE [5]. This design choice is also due to the above reason, where human feedback is typically associated with a short sequence of observations and actions instead of a single precise observation-action pair.
>
> >**Why would the moving window sampling help mitigate temporal inconsistency? I would imagine training a separate BT model on each temporal window would address inconsistency, at the expense of have potentially too many models. However, if you only use the windows to sample preference pairs, and train a BT model under the assumption that the reward model underlying all comparisons is the same, how can this address inconsistency?**
>
> Our moving window sampling only converts pairs of trajectories within each window into preference pairs. This makes sure that the reward model does not see any pairs that are far away from each other in terms of the temporal dimension. This ensures that, even if there is one reward model being trained on all data obtained from our moving window sampling, they do not assume that all comparisons are the same, since the training data only consists of comparisons between temporally close trajectories. Therefore, our method avoids temporal inconsistency from the beginning of training the reward model, that is, the training data. Moreover, our ablation studies in Section 5.2 show that by training the reward model without the moving window sampling to generate the training data, this model will have the exact same issue as the reviewer pointed out. It does not mitigate temporal inconsistency. These results show that our method of generating the training data is effective.

---

> ### Author Response · Authors · 2025-08-13
> **Response to Reviewer Pf2B (part 2)**
>
> (part2)
>
> >**For the ablation results in figure 6, is voting applied to generate the figure? Or is the data generated by different evaluators considered as a single evaluator for training the reward model here? I think the authors should clarify this. It seems to me that voting has the highest contribution to performance. If not the former, I think the authors should do an ablation on pref-GUIDE individual + voting with no temporal sampling and no preference range.**
>
> We would like to clarify that all results, including both Pref-GUIDE individual and Pref-GUIDE voting and their associated ablation studies (except one that we will outline as follows), are based on the average agent performance across all evaluators’ reward models. Pref-GUIDE voting does not train a single reward model by simply fusing all evaluators’ data. One ablation study in Figure 7 (i.e., Pref-GUIDE Voting combined) shows that training a single reward model by combining the data from all 50 evaluators into a single dataset performs worse than our methods. In our methods, even in Pref-GUIDE voting, the reward model for each evaluator is trained again based on the updated labels after querying the learned reward models from other evaluators (Section 4.3, last paragraph, Figure 2 (b), Algorithm Appendix A.3). We will clarify this in the paper. Therefore, our ablation studies make sense after this clarification to study the contributions of each module.
>
> >**How exactly are the rewards aggregated for training the policy?**
>
> Instead of aggregating rewards directly as in previous approaches, we aggregate the predictions of the naïve reward model (Pref-GUIDE Individual) from each human subject using voting, and use the normalized voting result as a soft label to re-train an improved reward model for each original reward model for each evaluator, which is our Pref-GUIDE Voting method. Details of the algorithm are provided in Appendix A.3.
>
> >**How many seeds did you use to repeat each experiment?**
>
> We use 5 seeds for all the experiments. We will add this to the paper.
>
> >**What RL algorithm did you use for the proposed method?**
>
> We use DDPG for the proposed method, Pref-GUIDE, following the hyperparameters and settings from GUIDE to ensure a fair comparison.
>
> *References*
>
> *[1] W Bradley Knox and Peter Stone. Interactively shaping agents via human reinforcement: The tamer framework. In Proceedings of the fifth international conference on Knowledge capture, pp. 9–16, 2009.*
>
> *[2] Garrett Warnell, Nicholas Waytowich, Vernon Lawhern, and Peter Stone. Deep tamer: Interactive agent shaping in high-dimensional state spaces. In Proceedings of the AAAI conference on artificial intelligence, volume 32, 2018.*
>
> *[3] James MacGlashan, Mark K Ho, Robert Loftin, Bei Peng, Guan Wang, David L Roberts, Matthew E Taylor, and Michael L Littman. Interactive learning from policy-dependent human feedback. In International conference on machine learning, pp. 2285–2294. PMLR, 2017.*
>
> *[4] Dilip Arumugam, Jun Ki Lee, Sophie Saskin, and Michael L Littman. Deep reinforcement learning from policy-dependent human feedback. arXiv preprint arXiv:1902.04257, 2019.*
>
> *[5] Lingyu Zhang, Zhengran Ji, Nicholas R Waytowich, and Boyuan Chen. Guide: Real-time humanshaped agents. In A. Globerson, L. Mackey, D. Belgrave, A. Fan, U. Paquet, J. Tomczak, and C. Zhang (eds.), Advances in Neural Information Processing Systems, volume 37, pp. 138959–138980. Curran Associates, Inc., 2024b. URL https://proceedings.neurips.cc/paper_files/paper/2024/ file/facf3192e99ce60c0ef5ed4067b72f68-Paper-Conference.pdf.*

---

> > ### Comment · Reviewer_Pf2B · 2025-09-10
> >
> > Thank the authors for the clarifications. I have no more questions.
> >
> > After some thoughts I found the temporal consistency idea interesting. You are intentionally avoiding contrasting/pairing trajectories from different windows, such that the reward model (hopefully) does not differentiate these pairs.

---

> > > ### Author Response · Authors · 2025-09-10
> > > **Thank you for your response**
> > >
> > > We are glad we addressed all your questions and concerns. Thank you!

---

### Review · Reviewer_9hr6 · 2025-08-22

**Summary Of Contributions:**

This paper presents Pref-GUIDE, a framework for human-in-the-loop continual RL. Pref-GUIDE addresses two key limitations from standard HITL with dense rewards: (1) the rewards from a single evaluator is not consistent, and (2) the rewards from multiple evaluators are not consistent. To address the first limitation, the paper proposes to convert dense rewards in a small time window to pairwise preferences and then fit a reward model to the preference labels (Pref-GUIDE Individual). To address the second limitation, the paper proposes to combine the individual preference reward models using a voting mechanism, and then fit an aggregate reward model to the voting-relabeled preferences (Pref-GUIDE voting). The paper evaluates Pref-GUIDE on three visual RL domains: Bowling, Find Treasure, and Hide and Seek 1v1. Compared to scalar human reward baselines (GUIDE) and standard sparse / dense reward RL baselines (DDPG), Pref-GUIDE shows strong robustness to noisy rewards, and often surpasses the performance of expert-designed dense rewards. This offers a promising pathway towards scalable human-in-the-loop RL.

**Audience:**

Yes

**Audience Explanation:**

Addressing the reward ambiguity problem in HITL is of immense interest to the machine learning and reinforcement learning communities.

**Broader Impact Concerns:**

The paper does not have ethical implications that warrant a Broader Impact Statement.

**Claims And Evidence:**

Yes

**Claims Explanation:**

The paper is motivated by concrete limitations of existing human-in-the-loop RL methods, namely, reward ambiguity. The proposed methods are based on clear insights, i.e. that individual evaluators have stationary preferences within a short time window, and that different evaluators can be reconciled via a voting mechanism. The empirical results convincingly demonstrate the effectiveness of the proposed methods.

**Requested Changes:**

1. The paper could benefit from a discussion of the limitations of the method, e.g., when does the time window assumption fail?
2. Some experiment details are omitted, e.g., how many seeds are in each experiment? What are the compute requirements? What are the exact dense rewards used in DDPG-Heuristic? Adding these details would help the reader better calibrate the results.

---

> ### Author Response · Authors · 2025-08-25
> **Response to Reviewer 9hr6**
>
> We thank the reviewer for your thoughtful comments. We would like to address all of your concerns and questions below with point responses.
>
> >***The paper could benefit from a discussion of the limitations of the method, e.g., when does the time window assumption fail?***
>
> The time window assumption may fail if the window size is too large to the extent that the human evaluator starts to show inconsistency in their guidance. Therefore, our experiments maintain a relatively short time window. All our experiments used the same parameter, showing that our method is not sensitive to the choice.
>
> For other discussions of potential limitations, we will add the following discussion of the limitations of the method in the conclusion section of our revised paper:
>
> “As for the limitation of our method, for the two design choices of Pref-GUIDE Individual: no-preference range and moving window, we assumed that the same set of parameters is sufficient to model different human evaluators. Though our results showed that the same set of parameters performed well,  these parameter selections may be further improved. For instance, more optimal subject-specific no-preference ranges and moving window sizes could potentially be inferred through a warm-up phase of human guidance. Moreover, for Pref-GUIDE Voting, more algorithm improvements can be studied to incorporate the human guidance from all subjects. Our work is also constrained to a single-task agent. Future work could extend Pref-GUIDE to multi-task settings.”
>
> >***Some experiment details are omitted, e.g., how many seeds are in each experiment? What are the compute requirements? What are the exact dense rewards used in DDPG-Heuristic? Adding these details would help the reader better calibrate the results.***
>
> Each experiment result was averaged over 5 seeds. We conducted the experiments using NVIDIA A100 and NVIDIA RTX A6000. However, our experiment can be run on a single desktop with a single GPU, such as NVIDIA RTX 4070. Since our experiments required a large number of training and evaluation with different human subjects, seeds, tasks, and ablation studies, we ran them on a cluster that is available to us. The dense rewards for Hide and Seek 1v1 and Find Treasure were defined based on the visibility of the treasure or hider and the distance between the agent and the target. We will add these details to the appendix section of the paper to clarify. Moreover, we will open-source all the code, dataset, and models upon publication.

---

### Decision · Action_Editor_GU8R · 2025-09-28

**Recommendation:** Accept as is

**Additional Comments:**

The paper proposes Pref-GUIDE, a framework for human-in-the-loop continual reinforcement learning that extends the earlier GUIDE method. Pref-GUIDE addresses two main challenges with human feedback: temporal inconsistency within a single evaluator’s responses and bias across multiple evaluators. To mitigate temporal inconsistency, the method generates preference pairs from short, adjacent time windows of scalar feedback and trains a reward model on these preferences. To handle evaluator bias, it trains separate reward models for each evaluator and aggregates them through a voting mechanism. The framework is evaluated in partially observable visual RL domains such as Bowling, Find Treasure, and Hide and Seek 1v1, showing improvements over GUIDE and DDPG baselines trained with sparse or expert-designed dense rewards.

The paper was evaluated by three reviewers. They appreciate the importance of addressing bias and inconsistency issues typical of contemporary human-in-the-loop RL methods and agree that the paper provides compelling empirical evidence of the benefits of the proposed approaches to address these issues. The reviewers initially raised several concerns with the paper as submitted. These included the need for a discussion of the limitations of the proposed methods; missing experimental details; and the concern that the method only achieves significant improvements in one of the evaluation domains.

The authors made a concerted effort to address each of the reviewers' concerns, both by clarifying misconceptions/misunderstandings on the part of the reviewers and by offering to revise the paper accordingly. As part of their official recommendation, all three reviewers acknowledge that the authors' responses have addressed their core questions/concerns.

**Audience:**

Yes

**Audience Explanation:**

As all three reviewers agree that the paper is highly topical in its focus on preference-based RL and that its consideration of human biases and inconsistencies will be of particular interest to many in the community.

**Claims And Evidence:**

Yes

**Claims Explanation:**

The reviewers emphasize that the paper provides clear empirical evidence that supports the effectiveness of the proposed methods, along with ablations that demonstrate the contributions of each of the components. Reviewer gwVU initially raised concerns about the lack of evidence in the form of per-trajectory time-series analysis to support the assumption that a trajectory that is rated positively may be rated negatively later. However, the authors resolved these concerns in their response, which the reviewer acknowledges in their final recommendation.